# Post-Marketing Surveillance of Nirsevimab: Safety Profile and Adverse Event Analysis from Spain’s 2023–2024 RSV Immunisation Campaign

**DOI:** 10.3390/vaccines13060623

**Published:** 2025-06-10

**Authors:** Pablo Estrella-Porter, Elisa Correcher-Martínez, Alejandro Orrico-Sánchez, Juan José Carreras

**Affiliations:** 1Medicina Preventiva, Hospital Clínico Universitario de Valencia, 46010 Valencia, Spain; 2Vaccine Research Department, Foundation for the Promotion of Health and Biomedical Research in the Valencian Region (FISABIO—Public Health), 46020 Valencia, Spain; elisa.correcher@fisabio.es (E.C.-M.); alejandro.orrico@fisabio.es (A.O.-S.); juanjo.carreras@fisabio.es (J.J.C.); 3Biomedical Research Consortium of Epidemiology and Public Health (CIBER-ESP), Instituto de Salud Carlos III, 28029 Madrid, Spain

**Keywords:** respiratory syncytial virus, nirsevimab, monoclonal antibodies, safety profile, infants

## Abstract

**Background:** Respiratory syncytial virus (RSV) poses a significant health burden in children, being the major cause of lower respiratory tract infection (LRTI), including bronchiolitis. During the 2023–2024 RSV season, Spain introduced nirsevimab, a monoclonal antibody for universal RSV prophylaxis in infants. This study reviews the safety of nirsevimab through post-marketing surveillance. **Material and Methods:** A descriptive pharmacovigilance study was made based on spontaneous reporting data of suspected adverse events (SAEs) from the Spanish Pharmacovigilance System for Medicinal Products for Human Use (SEFV-H) and industry reports. SAEs reported between September 2023 and May 2024 were extracted from the Spanish Pharmacovigilance Adverse Reactions Data (FEDRA) database. Cases were analysed by sex, age, severity, and SAEs classification using the Preferred Terms (PT) level of the Medical Dictionary for Regulatory Activities (MedDRA). Reporting rates were estimated based on immunization coverage and birth data. **Results:** Sixty-seven cases reported 141 SAEs, yielding an overall rate of 23.1 cases per 100,000 doses. Common events included rash (8.51%), drug ineffectiveness (7.09%), and pyrexia (7.09%). Serious events constituted 53.70% of reports, including two fatalities (3.00%). No new safety signals or unexpected risks, such as antibody-dependent enhancement (ADE), were identified. **Discussion:** SAEs reported peaked early in the campaign, reflecting heightened reporting in new immunization programs. The safety profile aligns with clinical trial findings and regulatory expectations, confirming nirsevimab’s benefit–risk balance. Continued pharmacovigilance is critical for maintaining public trust in RSV prophylaxis. Nirsevimab demonstrated a favorable safety profile during Spain’s initial universal RSV immunization campaign in infants, supporting its continued use in reducing RSV-related morbidity.

## 1. Introduction

Respiratory syncytial virus (RSV) infection represents a high burden of disease in infants, and in this population is the major cause of lower respiratory tract infection (LRTI), including bronchiolitis and pneumonia [1]. Previously, palivizumab was the main monoclonal antibody (mAb) against RSV used, and since the year 2000, Spain has funded its use in high-risk infant groups [2,3]. Palivizumab is administered as a 5-dose regimen, and systematic reviews have confirmed its effectiveness in preventing LRTIs, hospitalisations and ICU admissions, as well as its safety profile [3,4,5].

In October 2022, a new humanised IgG1κ mAb, nirsevimab, was approved in Europe for the prevention of RSV infection in children under 1 year of age, administered as a single-dose regimen [3,6]. This approval marked a significant advancement in RSV prophylaxis due to its simplified dosing schedule and broader potential for universal implementation.

Spain was one of the first countries to introduce nirsevimab into the national immunization program in the 2023–2024 RSV season. The campaign targeted infants under 6 months of age, born between April 2023 and March 2024, as well as high-risk infant groups [7]. Depending on the autonomous region, administration began between September 20th and 30th October 2023. Overall, most newborns during this period, including preterm infants and other high-risk groups, received nirsevimab in hospital settings prior to discharge. In contrast, infants born between April 1 and the start of the RSV season (catch-up group) predominantly received the prophylaxis in primary care facilities [8].

Following Spain’s first RSV immunisation campaign, which achieved a coverage of 91.9% [8], several studies from Spain estimated that nirsevimab is between 70 and 90% effective in preventing hospitalizations for RSV, depending on the study design [7,9,10,11,12,13,14]. Additionally, the monoclonal antibody has been estimated to be 76% effective in preventing primary care consultations for RSV LRTIs [14]. Overall, it could have prevented around 10.000 hospitalizations in infants under 1 year of age [15].

Initial clinical trials reported a favourable safety profile up to day 361 after nirsevimab administration [16], with the European Union Risk Management Plan (EU RMP) also not identifying any significant safety concerns in non-clinical studies [2]. Later updates to the EU RMP indicate that long-term safety is no longer considered a missing information category, as data from the MELODY (D5290C00004) and MEDLEY (D5290C00005) studies provided sufficient evidence supporting the safety profile of nirsevimab over two RSV seasons [17,18]. Known risks by the EU RMP, such as injection site reactions, rash, and pyrexia, do not alter the benefit–risk ratio. Immediate (type 1) hypersensitivity reactions, including anaphylaxis, are potential risks but do not require further characterisation. Similarly, the safety profile in children with comorbidities, such as heart or lung disease or prematurity, was comparable between those receiving nirsevimab and palivizumab [19], and a second dose in these children during their second RSV season maintained a favourable safety profile [20]. Although antibody-dependent enhancement (ADE) was initially a theoretical concern for monoclonal antibodies, clinical trials have not identified ADE as a relevant safety risk. No ADE events have been reported with RSV monoclonal antibodies, and a recent study monitoring children in their second RSV season after nirsevimab administration found no increase in disease severity compared to the placebo [2,16]. Other potential risks, such as immune complex disease, thrombocytopenia, and antiviral resistance, are not considered critical because they were either not observed in clinical trials or were found to be rare and non-serious and did not significantly impact the overall safety or efficacy of nirsevimab. As a result, planned pharmacovigilance activities have been adjusted accordingly, with a focus on post-marketing surveillance and the completion of previously ongoing studies [2].

The Spanish System for the Pharmacovigilance of Human Medicines (SEFV-H) is a decentralised network consisting of 17 regional pharmacovigilance centres, one for each autonomous community, coordinated by the Spanish Agency for Medicines and Health Products (AEMPS). Its main objective is to collect, evaluate, and record information on suspected adverse drug reactions, including those related to newly introduced medications such as nirsevimab. The system aims to identify previously unknown risks, detect changes in known risks, and provide continuous updates on medication safety [21].

There remains a critical need to evaluate safety outcomes in real-world settings. To date, all available safety data on nirsevimab have been derived from clinical trials, with no published real-world safety data. Spain, with over 277,000 doses administered and a robust pharmacovigilance system, provides an optimal setting to generate valuable post-marketing safety evidence. Accordingly, this study aims to describe the safety profile of nirsevimab through post-marketing surveillance, analysing suspected adverse events (SAEs) spontaneously reported to the SEFV-H during Spain’s 2023–2024 RSV immunisation campaign.

## 2. Materials and Methods

### 2.1. Study Design, Population and Period

A descriptive pharmacovigilance study based on spontaneous reporting data was conducted on reported cases of adverse events following nirsevimab administration in children under 1 year of age between 1 September 2023 and 31 May 2024, to allow for a two-month follow-up after the immunisation campaign concluded.

### 2.2. Case Definition

A suspected SAE following nirsevimab administration was defined as any reported event occurring after the receipt of the monoclonal antibody, irrespective of whether a causal relationship with the drug was established. It should be noted that a single case may include one or more SAEs.

### 2.3. Data Sources

SAEs after the administration of nirsevimab in children were obtained from the Spanish Pharmacovigilance Adverse Reactions Database (FEDRA) in the form of Individual Case Safety Reports (ICSR), which is managed by the SEFV-H, coordinated by the Spanish Agency for Medicines and Health Products (AEMPS). FEDRA operates as a passive reporting system, capturing information submitted spontaneously by health professionals, users, families, or caregivers; state health agencies; and pharmaceutical manufacturers, as well as industry reports [22]. All SAE data were coded using the Medical Dictionary for Regulatory Activities (MedDRA) according to its official guidelines [23]. Immunisation coverage data from the Spanish Ministry of Health and birth statistics from the National Statistics Institute were used to estimate notification rates.

### 2.4. Study Variables and Statistical Analysis

The study analysed demographic characteristics (sex and age), reporter category (health professionals, patients/consumers, marketing authorisation holders), date of nirsevimab administration, and severity assessment of SAEs. Reported events were classified using MedDRA version 27, categorised by System Organ Class (SOC) (Version: 27.0), and further described using Preferred Terms (PT) (Version: 27.0) [23].

A descriptive analysis was conducted for all study variables. SAEs were summarised at the MedDRA PTs level and grouped by SOC. Notification rates of SAEs per 100,000 doses were calculated using the total number of administered doses as the denominator, offering a standardised measure of the frequency of reported events.

### 2.5. Ethical Considerations

The results, discussion, and conclusions of this paper show the author’s point of view, and they do not represent in any way the position of the SEFV-H or the AEMPS regarding this issue. All data used in this study were anonymised by the SEFV-H before being loaded into the FEDRA database, from which they were accessed for analysis. As SAE reporting is part of standard pharmacovigilance practices, informed consent was not required.

## 3. Results

During the 2023–2024 RSV immunisation campaign in Spain, a total of 67 cases reported a total of 141 SAEs following nirsevimab administration, giving an average of 2.10 SAEs per case and an overall notification rate of 23.1 cases per 100,000 doses administered and 48.6 SAEs per 100,000 doses. These events encompassed a total of 86 distinct PTs across 15 different SOCs. Case reports were predominantly from males (62.7), and the mean age was 3.24 months. While 79.1% were infants, 20.9% were newborns (Table 1). The notification rate was 28.1 per 100,000 doses administered in males and 17.8 per 100,000 doses in females. A total of 30.0% of SAEs occurred in the first 48 h after the administration, and the median latency from the first dose to the onset of SAEs was 4.00 days (range 0–112.00 days). Of all SAEs, 94% were directly submitted to the SEFV-H, primarily by physicians (52.2%), followed by pharmacists (20.3%) and nurses (18.8%). Notifications were almost evenly distributed between in-hospital settings (44.9%) and out-of-hospital settings (43.5%). The majority of cases (40.3%) involved a single SAE, while 32.8% reported two SAEs, and 26.9% involved three or more SAEs.

The monthly distribution of SAEs (Figure 1) showed a higher concentration in the initial months of the campaign. October 2023 accounted for the largest share of cases (41.8%, 28 cases), followed by November 2023 and January 2024 (17.9% each, 12 cases). December 2023 represented 14.9% of the cases (10 cases), whereas February 2024 and March 2024 had significantly fewer cases, with 1.5% (1 case) and 6.0% (4 cases), respectively.

A total of 36 cases (53.7%) were classified as serious, and 21 (31.3%) required hospitalisation, while 7 (10.4%) had prolonged hospital stays (Table 2). There were two reported deaths, one of which occurred after prior hospitalisation (1.5% each). Regarding patient outcomes at the time of notification (Table 3), 55.2% of cases achieved full recovery, while 26.9% were still in the recovery process and 6.0% showed no signs of recovery.

In addition, 26 documented baseline medical conditions were reported among the cases, with prematurity (11.5%) being the most frequent (Table 4).

The most frequently reported SAEs (Table 5) were categorised under the SOC general disorders and administration site conditions, accounting for 26.95% of all cases. Within this SOC, the most common PTs were drug ineffectiveness and pyrexia (7.09%). The SOC Infections and Infestations contributed 18.44% of cases, mainly driven by RSV bronchiolitis (6.38%). Skin and subcutaneous tissue disorders SOC represented 17.73%, with rash being the most frequent PT (8.51%). Other notable SOCs included nervous system disorders and respiratory, thoracic, and mediastinal disorders, each contributing 7.80%, highlighting events such as lethargy (1.42%) and acute respiratory failure (2.13%). Gastrointestinal disorders accounted for 5.67%, with vomiting (2.13%) as the most notable PT. These results underscore the predominance of general site reactions and infections as the main categories of reported adverse events during the study period.

Notably, upon reviewing all spontaneous notifications received during this period, none of the potential risks outlined in the EU RMP, such as immune complex disease, thrombocytopenia (type III hypersensitivity), ADE, antiviral resistance, or immediate (type 1) hypersensitivity reactions (including anaphylaxis), were identified.

## 4. Discussion

During the 2023–2024 RSV immunisation campaign in Spain, the overall notification rate was 23.1 cases per 100,000 doses. The predominant SAEs reported through the SEFV-H, such as injection site reactions and pyrexia, were expected and did not alter the overall benefit–risk profile of the drug, which is consistent with clinical expectations and aligns with findings from other studies where these events were also reported as the most common, generally mild, and short lived [24]. Importantly, the absence of reported cases of ADE, thrombocytopenia, or severe hypersensitivity reactions confirms previous clinical trial findings [25], even though the EU RMP outlines them as potential theoretical risks [2]. The safety profile of nirsevimab as reported aligns with the medical products’ data sheet, and no additional adverse reactions were identified [2,26].

Clinical trial data further reinforce this favourable profile, with pooled analyses from phase 2b, 2/3, and 3 trials (total >3000 nirsevimab recipients) showing that overall adverse-event rates, predominantly mild or moderate, were similar to placebo or palivizumab, with serious adverse events (<1%) and fatalities (<1%) deemed unrelated to nirsevimab administration [27]. A recent Q1 2025 comprehensive review also noted that rash, fever, and injection-site reactions were the most frequent adverse events within 7–14 days post-dose, further supporting the safety profile observed in real-world surveillance, despite the limited post-marketing exposure to date [28].

International post-marketing experience has further reinforced these findings. In Western Australia, an active SMS-based surveillance of over 4300 infants (April–July 2024) reported that 27.5% experienced any reaction, most commonly injection-site redness, fatigue, and fever, with rash less frequent, and only 1.5% sought medical evaluation for an adverse event following immunization (AEFI), with no serious events identified [29]. Likewise, Italy’s Valle d’Aosta universal program (May 2023–February 2024) observed only mild, transient effects (lasting 1–2 days), including fever in 6.5%, local injection-site reactions in 4%, and inconsolable crying in 0.4% of infants [30]. These data suggest that nirsevimab’s tolerability is consistent across diverse health systems and immunisation settings. SAEs reported during the first nirsevimab immunisation campaign in Spain showed a peak in notifications early in the campaign, mirroring patterns previously described in the rollout of other immunisation programs, where early heightened awareness among professionals and parents typically leads to increased reporting [22,31]. This trend gradually declined towards the end of the season. Overall, the safety findings were consistent with the safety profile outlined in the EU RMP [2,26].

It is important to emphasise that these data come from spontaneous notifications, which do not represent all actual events and do not necessarily imply causality. The reports reflect a temporal association rather than a confirmed causal relationship, and their primary goal is to detect potential unknown risks.

According to nirsevimab’s summary of product characteristics [26], the rash is the most frequently reported adverse reaction, typically occurring within 14 days post-dose, while pyrexia and injection site reactions generally appear within seven days after the dose [2]. In this study, rash presented a median latency of 6.0 days, 2.0 days for pyrexia, and 2.0 days for injection site reactions. These latency distributions are consistent with both the product label and international post-marketing findings [27,28], supporting the reliability of the observed safety profile.

Given the estimated efficacy of 74.5% in clinical trials [6] and 71.0% effectiveness in the prevention of RSV hospitalisations in the catch-up group in Spain [32], it is expected that some cases of RSV infection or bronchiolitis would still occur despite nirsevimab administration. The reported cases of bronchiolitis (6.38%) and drug ineffectiveness (7.09%) in this analysis are in line with these efficacy estimates, suggesting that while nirsevimab offers substantial protection, a residual risk of infection remains. The occurrence of bronchiolitis cases despite prophylaxis does not necessarily indicate a safety issue but rather reflects the limitations inherent to any preventive intervention, especially in real-world settings where clinical conditions and compliance may vary. These findings are congruent with other real-world reports [27,28], reinforcing the importance of contextualising SAEs within known vaccine effectiveness parameters.

When discussing parental acceptance of nirsevimab after birth, a French prospective longitudinal cohort study identified safety concerns, specifically the balance between parents’ desire to protect their infant and their worries about insufficient long-term data and potential side effects, as key determinants of reluctance [33]. Consequently, robust safety and adverse-event data are critical to addressing hesitancy and facilitating informed, shared decision making with patients and their families.

The high immunization coverage achieved during this campaign and the estimated notification rate of 23.1 cases per 100,000 administrations further support the safety of nirsevimab. Interestingly, there was a nearly two-fold higher notification rate among male infants (14.5 per 100,000 doses) compared to female infants (8.6 per 100,000 doses), which aligns with the pattern observed in literature for that population age group [34]. Since adverse events are often reported in children under 2 years of age [35], the proactive involvement of healthcare professionals and patients in reporting is essential to maintaining pharmacovigilance systems and ensuring the safety of new immunisation programs. In this way, continued monitoring and international collaboration will be essential to ensure early detection of rare or long-term adverse events and to sustain public confidence.

## 5. Conclusions

While the report of severe events underscores the need for continued vigilance, the overall safety profile of nirsevimab remains reassuring and supports its continued use as a preventive measure against RSV. This campaign provides a solid foundation for future RSV immunisation efforts and highlights the importance of robust pharmacovigilance to maintain public trust in new health interventions.

With this review of spontaneous notifications reported to the SEFV-H during the first season, no new safety concerns associated with nirsevimab administration in the paediatric population were identified beyond those already described in the product’s Summary of Product Characteristics, which aligns with post-marketing data from Italy [30] and Western Australia [29].

## Figures and Tables

**Figure 1 vaccines-13-00623-f001:**
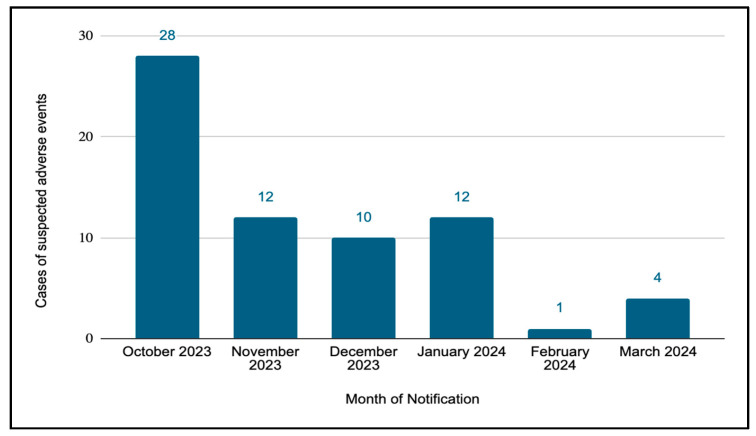
Monthly distribution of cases of adverse events by date of notification following nirsevimab administration during the 2023–2024 RSV immunisation campaign in Spain.

**Table 1 vaccines-13-00623-t001:** Demographics and baseline characteristics.

Characteristic		Total (*n* = 67)
Age at dosing	Mean age in months (SD)	3.24 (3.25)
	Median age in months (IQR)	2.03 (1.0, 5.1)
Age group, *n* (%)	Infant	53 (79.1%)
	Newborn	14 (20.9%)
Sex, *n* (%)	Female	25 (37.3)
	Male	42 (62.7)
SAEs per case, *n* (%)	1	27 (40.3%)
	2	22 (32.8%)
	3	12 (17.9%)
	4	2 (3.0%)
	5	2 (3.0%)
	7	1 (1.5%)
	8	1 (1.5%)

SAEs: suspected adverse events.

**Table 2 vaccines-13-00623-t002:** Level of severity of cases of adverse events reported during the 2023–2024 RSV immunisation campaign in Spain.

Category	Cases	Percentage (%)	Notification Rate (Per 100,000 Doses)
Non-serious	31	46.3%	10.7
Life-threatening, medically significant condition	1	1.5%	0.3
Life-threatening, requires hospitalisation	8	11.9%	2.8
Life-threatening, requires hospitalisation, medically significant condition	1	1.5%	0.3
Life-threatening, requires hospitalisation, results in persistent/significant disability	1	1.5%	0.3
Requires hospitalisation	10	14.9%	3.4
Prolongs hospitalisation	7	10.4%	2.4
Medically significant condition	6	9.0%	2.1
Fatal	1	1.5%	0.3
Fatal, requires hospitalisation	1	1.5%	0.3
**Total**	67	100.0%	23.1

**Table 3 vaccines-13-00623-t003:** Distribution of cases of adverse event outcomes reported in the SEFV-H during the 2023–2024 RSV immunisation campaign in Spain.

Case Outcome	Number of Cases	Percentage	Notification Rate (Per 100,000 Doses)
Recovered	37	55.2%	12.8
Recovering	18	26.9%	6.2
Not recovered	4	6.0%	1.4
Fatal	2	3.0%	0.7
Unknown	6	9.0%	2.1
Total	67	100.0%	23.1

**Table 4 vaccines-13-00623-t004:** Baseline medical history from the cases of adverse events reported in the SEFV-H during the 2023–2024 RSV immunisation campaign in Spain.

Medical History	Number of Notifications
Premature neonate	3
Bronchiolitis	2
Low birth weight neonate	2
Nephrogenic anaemia	1
Multiple congenital abnormalities	1
Central venous catheterization	1
Patent ductus arteriosus	1
COVID-19	1
Ventricular septal defect	1
Foetal malnutrition	1
Bronchopulmonary dysplasia	1
Epileptic encephalopathy	1
Congenital pulmonary valve stenosis	1
Cystic fibrosis	1
Gastrostomy	1
Pneumothorax	1
Foetal growth restriction	1
Neonatal sepsis	1
Sturge–Weber syndrome	1
Foetal distress syndrome	1
Congenital nephrotic syndrome	1
VACTERL syndrome	1
**Total**	**26**

**Table 5 vaccines-13-00623-t005:** Distribution of SAEs reported in the SEFV-H during the 2023–2024 RSV immunisation campaign in Spain by MedRA Preferred Terms (PT).

Preferred Terms (PT) **	Number of SAEs	Percentages (%)	Notification Rate (Per 100,000 Doses)	Median Latency (in Days) After the First Dose
Rash	12	8.51%	4.1	6.0
Drug ineffective	10	7.09%	3.4	52.5
Pyrexia	10	7.09%	3.4	2.0
Respiratory syncytial virus bronchiolitis	9	6.38%	3.1	64.0
Bronchiolitis	4	2.84%	1.4	4.0
Acute respiratory failure	3	2.13%	1	6.0
Urticaria	3	2.13%	1	14
Vomiting	3	2.13%	1	1.0
Metabolic acidosis	2	1.42%	0.7	21.8
Diarrhoea	2	1.42%	0.7	3.0
Maculo-papular rash	2	1.42%	0.7	3.5
Gastroenteritis	2	1.42%	0.7	37.0
Upper respiratory tract infection	2	1.42%	0.7	13.0
Lethargy	2	1.42%	0.7	0.8
Skin reaction	2	1.42%	0.7	3.5
Oxygen saturation decreased	2	1.42%	0.7	2.0
Injection site urticaria	2	1.42%	0.7	1.0

** Only preferred terms (PTs) with more than one SAE notification were included in this table. SAEs: suspected adverse events

## Data Availability

Data are contained within the article.

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
