# Peer review of "Post-Marketing Surveillance of Nirsevimab: Safety Profile and Adverse Event Analysis from Spain’s 2023–2024 RSV Immunisation Campaign"

_vaccines, 2025, doi:10.3390/vaccines13060623_

Round 1
Reviewer 1 Report
Comments and Suggestions for Authors
Post-Marketing Surveillance of Nirsevimab: Safety Profile and 2 Adverse Event Analysis from Spain’s 2023–2024 RSV 3 Immunisation Campaign by Pablo Estella-Porter et al.
The article analyses spontaneous reporting of adverse events to nirsevimab, a humanized monoclonal antibody for RSV prophylaxis in infants, after its introduction in the 2023-24 season in Spain. 23 cases of adverse effects per 100,000 doses were reported, with roughly half considered serious including 2 death cases. However, antibody-dependent enhancement of RSV disease was not observed and nirsevimab's benefit-risk balance was confirmed in agreement with previous evaluations.
The Authors conclude that nirsevimab is safe and no novel adverse effect beyond the ones reported were found.
English: no language issues were found.
I consider this study worth publishing so that the rate of adverse effects is known and may serve as a comparison for future and past studies on the same subject.
Author Response
Thank you very much for taking the time to review this manuscript.
We appreciate your positive assessment of the manuscript and are pleased that you consider the study a valuable contribution for future comparisons of nirsevimab safety data.
Reviewer 2 Report
Comments and Suggestions for Authors
The research article is very well-written, interesting, and extensive. I recommend it for publication in the Vaccines journal.
Author Response
Thank you very much for taking the time to review this manuscript.
We appreciate your positive assessment of the manuscript.
Reviewer 3 Report
Comments and Suggestions for Authors
The information provided by the article is very important and relevant.
However could the authors be so kind and better explain, for the benefit of the readers how the vaccination schedule was performed?
Were the SAE linked to the one or two doses of the vaccine provided? Was there any catch-up program? Did infants that received palivizumab receive also nirsevimab?
Author Response
Thank you very much for taking the time to review this manuscript.
Please find the detailed responses below and the corresponding revisions highlighted changes in the re-submitted file.
Comment 1: The information provided by the article is very important and relevant.
However could the authors be so kind and better explain, for the benefit of the readers how the vaccination schedule was performed?
Response 1:
Thank you for pointing this out. We agree that a clearer description of the vaccination schedule enhances reader understanding. Accordingly, we have expanded the Introduction (page 2, paragraphs 1–2) to provide the context of RSV immunisation with palivizumab prior to the introduction of nirsevimab, detail the dosing regimens and it is complimented by paragraph 3 that highlights the target populations, immunization settings (hospital versus primary care), time frame, and catch-up cohort for nirsevimab administration.
Comment 2: Were the SAE linked to the one or two doses of the vaccine provided? Was there any catch-up program? Did infants that received palivizumab receive also nirsevimab?
Answer 2:
Thank you for this important comment.
As described in the revised Introduction (page 2, paragraphs 1–2), nirsevimab was administered as a single‐dose regimen during the 2023–2024 season. Consequently, all SAEs reported in our study occurred following that single nirsevimab dose; no dual‐dose schedule was used.
The catch-up cohort is defined in paragraph 3 of the introduction as "infants born between April 1 and the start of the RSV season (catch-up group) ) predominantly received the prophylaxis in primary care facilities".
The revised Introduction (page 2, paragraphs 1–2) now explains that palivizumab’s five‐dose series was replaced by a single dose of nirsevimab, and describes how this change was incorporated into the 2023–2024 immunisation program.